# Interface-based tuning of Rashba spin-orbit interaction in asymmetric oxide heterostructures with 3$d$ electrons

Weinan Lin[1,2,12], Lei Li[3,4,12], Fatih Doğan[5,6], Changjian Li[1], Hélène Rotella[7], Xiaojiang Yu[8], Bangmin Zhang[1], Yangyang Li[1], Wen Siang Lew [2], Shijie Wang[9], Wilfrid Prellier [7], Stephen J. Pennycook[1], Jingsheng Chen [1], Zhicheng Zhong[3,4,10], Aurelien Manchon [6] & Tom Wu[11]

The Rashba effect plays important roles in emerging quantum materials physics and potential spintronic applications, entailing both the spin orbit interaction (SOI) and broken inversion symmetry. In this work, we devise asymmetric oxide heterostructures of $LaAlO_3$//$SrTiO_3$/$LaAlO_3$ (LAO//STO/LAO) to study the Rashba effect in STO with an initial centrosymmetric structure, and broken inversion symmetry is created by the inequivalent bottom and top interfaces due to their opposite polar discontinuities. Furthermore, we report the observation of a transition from the cubic Rashba effect to the coexistence of linear and cubic Rashba effects in the oxide heterostructures, which is controlled by the filling of Ti orbitals. Such asymmetric oxide heterostructures with initially centrosymmetric materials provide a general strategy for tuning the Rashba SOI in artificial quantum materials.

[1] Department of Materials Science and Engineering, National University of Singapore, Singapore 117575, Singapore. [2] Division of Physics and Applied Physics, School of Physical and Mathematical Sciences, Nanyang Technological University, Singapore 637371, Singapore. [3] CAS Key Laboratory of Magnetic Materials and Devices Ningbo Institute of Materials Technology and Engineering, Chinese Academy of Sciences, 315201 Ningbo, Zhejiang, China. [4] Zhejiang Province Key Laboratory of Magnetic Materials and Application Technology, Ningbo Institute of Materials Technology and Engineering, Chinese Academy of Sciences, 315201 Ningbo, Zhejiang, China. [5] College of Engineering and Technology, American University of the Middle East, 15453 Eqaila, Kuwait. [6] Physical Sciences and Engineering, King Abdullah University of Science and Technology, Thuwal 23955-6900, Saudi Arabia. [7] Laboratoire CRISMAT, ENSICAEN, CNRS UMR 6508, 6 Boulevard du Maréchal Juin, 14050 Caen, France. [8] Singapore Synchrotron Light Source, National University of Singapore, Singapore 117575, Singapore. [9] Institute of Materials Research and Engineering, Singapore 117602, Singapore. [10] China Center of Materials Science and Optoelectronics Engineering, University of Chinese Academy of Sciences, 100049 Beijing, P.R. China. [11] School of Materials Science and Engineering, University of New South Wales (UNSW), Sydney, NSW 2052, Australia. [12] These authors contributed equally: Weinan Lin, Lei Li. Correspondence and requests for materials should be addressed to J.C. (email: msecj@nus.edu.sg) or to Z.Z. (email: zhong@nimte.ac.cn) or to A.M. (email: aurelien.manchon@kaust.edu.sa) or to T.W. (email: tom.wu@unsw.edu.au)

Manipulation of the spin degree of freedom can be expanded to various nonmagnetic materials by the Rashba effect, where both the spin–orbit interaction (SOI) and broken inversion symmetry are required[1–4]. The phenomenon is generally observed at the surface of heavy metals and in asymmetric two-dimensional quantum wells, where inversion symmetry is broken naturally[3,5]. Since the Rashba effect plays a key role not only in fundamental physics but also for potential applications, there has been considerable effort devoted to exploring and/or maximizing the Rashba effect in a wide range of materials[6]. In principle, an external electric field can break and tune the inversion symmetry of the materials studied[7,8], which is also the mechanism of the celebrated spin transistor[9]. However, such a technique is limited by the magnitude and the screening length it can achieve[10–12]. As a result, much attention has been paid to exploring structures with intrinsically broken inversion symmetry, such as ferroelectrics and polar semiconductors[13–16]. Recently, the effort has been extended to manipulate the geometric environment of orbitals to achieve a giant Rashba effect[6,17,18].

Complex perovskite oxides $ABO_3$ show great potential for exploiting the Rashba effect[6,19] because their multiple degrees of freedom, i.e., charge, spin, orbital, and lattice, are entangled with one another[20,21]. In heterostructures of perovskite oxides $ABO_3$ and $A'B'O_3$, two different interfaces, i.e., AO-B'O$_2$ and BO$_2$-A'O, in the [001] direction can be introduced[22,23]. SrTiO$_3$ (STO), which has $3d$ electrons, is a prototypical example (Fig. 1a). However, there is no spin splitting in the $3d$ electron band for the centrosymmetric STO, and strain and SOI can lift the degeneracy of only the $3d$ orbitals (Fig. 1b). Spin splitting requires breaking the inversion symmetry of the structure, for example, when STO forms a heterointerface with (LaO)$^+$ interface due to the polar discontinuity of LAO[23–27]. Such a significant Rashba effect in the system has led to the discovery of significant spin–charge conversions[28–30] and the perspective of various exotic properties, such as skyrmion[31,32], topological superconductivity[33], and intrinsic spin Hall effect[34]. In this heterostructure, the inversion symmetry at the interface is naturally broken, and the band structure is significantly modified owing to the confinement effect and other interface-related effects[20,22].

As illustrated in Fig. 1a (right), we propose in this work that broken inversion symmetry can be manipulated by creating two unequal interfaces in LAO//STO/LAO oxide heterostructures, which makes use of their opposite polar discontinuities, i.e., (AlO$_2$)$^-$-(SrO) versus (TiO$_2$)-(LaO)$^+$. In this scheme, the potential difference between these two unequal interfaces produces a built-in electric field to break the inversion symmetry of STO[35]. Compared to the well-studied LAO/STO heterostructure (Supplementary Figs. 1 and 2), where the inversion symmetry is naturally broken at the interface, this designed approach is capable of modulating the broken inversion symmetry via tuning the intermediate STO layer thickness. Combining density functional theory (DFT)-based tight-binding calculations with weak antilocalization (WAL) measurements, the transition from the pure cubic Rashba term to its coexistence with the linear term is identified as a function of carrier filling in Ti orbitals. Our work demonstrates a general platform for exploring Rashba SOI physics in interface-asymmetric heterostructures with initially centrosymmetric materials.

## Results

**Tuning of Rashba effect in the LAO//STO/LAO heterostructures.** The band structure of STO with a centrosymmetric structure and compressive strain from LAO was obtained from DFT-based tight-binding calculation (Fig. 1b). Owing to the biaxial strain in the (001) plane, the degeneracy of Ti $t_{2g}$ is lifted with the $d_{xy}$ orbital located above the $d_{yz/xz}$ orbitals. When considering the SOI, the degeneracy of the $d_{yz/xz}$ band is further lifted. Nevertheless, owing to the inversion symmetry of STO, the energy band is doubly degenerate in the entire Brillouin zone, which does not result in spin splitting (indicated by the green arrows in Fig. 1b). As proposed above, the feature can be achieved by introducing two unequal interfaces between LAO and STO, which breaks the inversion symmetry of STO, as confirmed by our calculation (Fig. 1c). The resulting Rashba effect lifts the double degeneracy of the energy band away from the Γ point, i.e., resulting in spin splitting of the $3d$ orbitals (green arrows in Fig. 1c).

Furthermore, the linear and cubic Rashba terms, where the spin splitting is linearly and cubically proportional to the momentum **k** (inset of Fig. 1d), respectively, are predicted to coexist in the $t_{2g}$ multi-orbitals. These two types of Rashba effects result in different spin configurations in each Fermi surface and may further lead to different spin-related properties[36–38]. Importantly, the characteristics of the Rashba effect can be controlled by the filling of carriers at the Ti sites. Figure 1d shows the spin splitting energy as a function of the carrier concentration at each Ti atom for the cubic and linear Rashba effects. In the STO with inversion symmetry broken, the linear Rashba effect emerges when the carrier concentration reaches approximately 0.01 e/Ti. We should note that this critical carrier concentration is an intrinsic property of STO and determined by its electronic band structure. We hypothesize that the asymmetric interfaces of the LAO//STO/LAO structures can be validated by characterizing the Rashba effect, which might be tuned by the carrier concentration in the STO layers. Such asymmetric multilayers consisting of initially centrosymmetric materials provide a general platform for investigating the Rashba SOI in engineered heterostructures.

It is important to note that, although the existence of the linear Rashba effect was theoretically predicted in oxide herterostructures with Ti $3d$ electrons[38–40], so far only the cubic Rashba effect has been reported in transport studies[25,41,42]. One possible explanation is that the electrons from the $d_{xy}$ orbital are localized at the interface and do not contribute to charge transport[43]. In this work, we envision that the itinerant nature of the electrons in the asymmetric LAO//STO/LAO heterostructures will enable us to explore the linear Rashba effect and its coexistence with the cubic Rashba effect will be modulated via carrier filling in Ti orbitals.

**Heterostructure synthesis and scanning transmission electron microscopy (STEM) characterization.** The oxide heterostructures were grown on treated LAO substrates using pulsed laser deposition (PLD). The thickness and the termination of the LAO and STO layers are precisely controlled via monitoring reflection high-energy electron diffraction (RHEED) (see Supplementary Fig. 4a for a typical heterostructure with 15 uc STO). As discussed in the previous section, as a result of the polar nature of LAO, such oxide heterostructures are designed to be asymmetric: the (AlO$_2$)$^-$–SrO interface between the LAO substrate and the STO layer is presumably $p$ type[23,44], while the top TiO$_2$–(LaO)$^+$ interface is $n$ type. However, as we will discuss below, factors like cation intermixing at the interfaces make the actual heterostructure deviate from the design although the asymmetric nature is retained.

To confirm the high quality of the films, X-ray reflectivity experiments were conducted, and a typical result is shown in Supplementary Fig. 4b. From the simulation of the reflectivity data, we derived the thicknesses of each layer in the

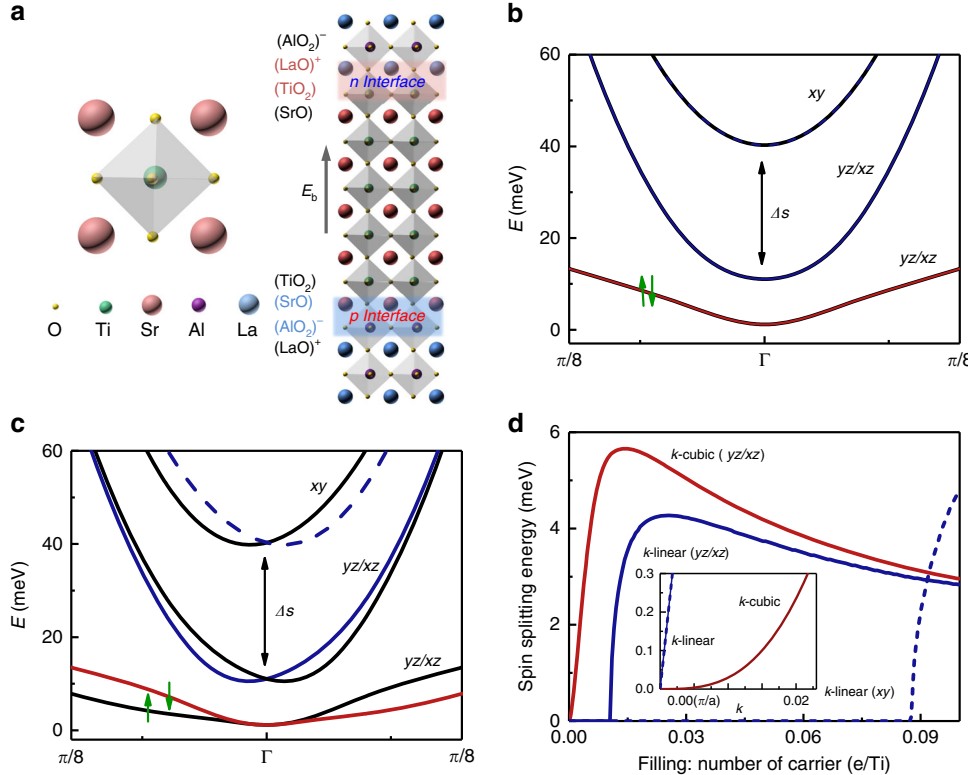

**Fig. 1** Spin splitting in the LAO//STO/LAO heterostructure. **a** Schematic structure of STO (left) and the asymmetric LAO//STO/LAO heterostructure (right) with a large built-in electric field $E_b$. **b** Band structure of $t_{2g}$ orbitals in the centrosymmetric STO layer. With compressive strain and spin–orbit interaction (SOI), the degenerate $t_{2g}$ orbitals are lifted with a splitting energy $\mathbf{\Delta_s}$. **c** Band structure of $t_{2g}$ orbitals with inversion symmetry broken as in the LAO//STO/LAO heterostructure. The degeneracy of the $d_{xy}$ and $d_{yz/xz}$ orbital is further lifted by the SOI. **d** Spin splitting energy of the $t_{2g}$ orbitals as a function of electron filling. The inset shows the $k$-linear and the cubic spin splitting of the corresponding orbitals

heterostructure, which are consistent with the RHEED results. By fitting the fringe oscillations of the high-resolution X-ray theta-2theta measurements, we extracted the out-of-plane $c$ lattice parameter of the STO layer. Interestingly, it was found that $c$ decreases from 3.946 to 3.895 Å when the STO thickness decreases in the series of heterostructures (see Supplementary Fig. 4d). Such a feature is ascribed to the electrostriction effect due to the built-in electric field[45], which is supported by our DFT calculations (see Supplementary Note 3).

To further characterize the strain state of the STO layer and, especially, the interface atomic structures, aberration-corrected STEM was used. A typical LAO//STO/LAO structure with 20 uc STO was characterized. As shown by the high-angle annular dark-field (HAADF)-STEM images in Fig. 2, epitaxial and coherent growth of the STO layer is confirmed. The cross-section HAADF-STEM image of the LAO//STO/LAO hetero-structure and the strain components parallel ($\varepsilon_{xx}$) and perpendicular ($\varepsilon_{yy}$) to the interface are presented in Supplementary Fig. 5, which further supports the coherent growth of the heterostructure. Cross-sectional elemental mapping was also obtained by atomically resolved STEM-energy-dispersive X-ray spectroscopy, from which we can acquire information on the local composition around the interfaces in such heterostructures. Similar to reports on the STO/LAO interface[44,46,47], cation intermixing was observed at both interfaces in the LAO//STO/LAO heterostructure. More importantly, we found that the intermixing of the top and bottom interfaces are different; it spans approximately 1 uc for the bottom interface and 2 uc for the top interface, as indicated in the overlay images in Fig. 2. This variation in the cation mixing has been discussed as one of the key differences of

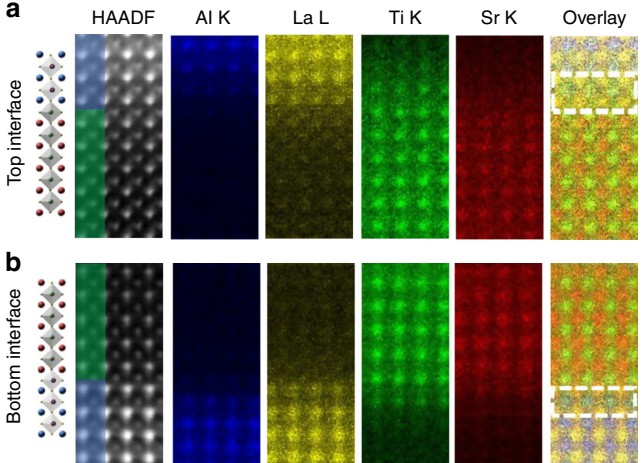

**Fig. 2** Atomic structural characterization of the two interfaces in the LAO//STO/LAO heterostructure. **a**, **b** show the energy-dispersive X-ray spectroscopic (EDS) mappings of the top and bottom interfaces, respectively. From left to right: schematic structures, high-angle annular dark-field image, Al K, La L, Ti K, and Sr K EDS integrated signal maps, and the combined elemental maps of all cations

$n$-type and $p$-type interfaces[44]. It should be noted that the cation intermixing at the bottom interface results in a combination of $(AlO_2)^-$-SrO and $(LaO)^+$-$TiO_2$ configurations. This atomic reconstruction can be regarded as a response to the polar discontinuity due to an energy gain compared to that from

generating the hole-doped interface, which was reported in other similar oxide heterostructures[48,49]. Although the atomic structures of the interfaces deviate from the ideal case (Fig. 1a), the LAO//STO/LAO heterostructure remains asymmetric.

**Thickness-dependent density of 3d electrons.** Hall effect measurements were performed to obtain the carrier density of the LAO//STO/LAO heterostructures. Figure 3a shows the typical Hall measurements for the structure with 8, 20, and 30 uc STO, from which the sheet carrier density was calculated. It was found that the sheet carrier concentration ($n_H$) increased progressively from $1.07 \times 10^{14}$ to $1.92 \times 10^{14}$ cm$^{-2}$ with decreasing STO layer thickness, as shown in Fig. 3b. It is important to note that the carrier concentration is higher in the heterostructures with thinner STO layers. The corresponding averaged-filling electrons per Ti in the heterostructure exhibits a similar trend as a function of the STO thickness (inset of Fig. 3b). In the rough estimation, the carriers are assumed to be homogeneously doped in the entire STO layer, which does not reflect the local deviation of the electron filling across the STO layer. As expected from the designed LAO/STO/LAO structure with two non-equivalent interfaces, the electron distribution across the whole STO layer should show a gradual increase from the $p$-type interface to the $n$-type interface. Furthermore, the localized electrons near the interfaces may have an important influence on the internal electric field and impact on the Rashba splitting. Nevertheless, the carrier density estimated from the Hall effect measurement exhibits a consistent trend of dependence on the STO layer thickness. When the STO layer in the heterostructure increases from 8 uc to 60 uc, the number of electrons per Ti decreases from 0.038 to 0.003. This range of electron filling is smaller than that in the extensively investigated LAO/STO structures where the induced electrons are mainly located around the interface.

In such heterostructures, both the electrons due to the polar LAO layer and those doped by the defects introduced during growth may contribute to the conduction. If the carriers were introduced mainly by the growth-related defects, the sheet carrier concentration ($n_s$) with a bulk origin would increase with the STO thickness, which is opposite to our observation (inset of Fig. 3b). Thus defects are not the dominant source of conducting carriers. Furthermore, a control experiment was conducted on a heterostructure with amorphous La$_{0.7}$Sr$_{0.3}$MnO$_3$ (a-LSMO) as the capping layer, i.e., LAO/STO/a-LSMO, which was prepared under the same conditions as the LAO//STO/LAO heterostructures except that the top LSMO layer was deposited at room temperature. The absence of measurable conductivity in such a LAO//STO/a-LSMO structure indicates that the polarity of the LAO plays an essential role in introducing the itinerant carriers[35], which may be achieved by influencing the formation energy of the oxygen vacancy.

**Evolution of SOI in LAO//STO/LAO heterostructures.** As quantum corrections to the conductance, both weak localization (WL) and WAL can be present at low temperatures in oxide heterostructures. Because these effects are sensitive to an external magnetic field, magnetotransport measurements can provide insights into the nature of SOI[50]. Figure 4 shows the magnetoresistance (MR) of a heterostructure with 30 uc STO inserted between the LAO layers. The negative MR at 20 and 30 K is a signature of WL, while with decreasing temperature, a cusp emerges around the zero field and broadens progressively. Such MR features at low temperatures are manifestations of WAL and are indicative of the presence of a strong SOI in the systems[25,41].

By varying the thickness of the STO layer in the heterostructures, magnetotransport of the LAO//STO/LAO heterostructures can be systemically tuned. As shown for the 2 K MR

data in Fig. 5a, the zero-field cusp shrinks with increasing STO layer thickness, and finally, a negative MR emerges with the high magnetic field at the thickest STO layer of 60 uc. This indicates a crossover from the WAL to the WL regime with increasing STO thickness. In the WL, the phase coherence of itinerant electrons is destroyed mainly by inelastic scattering[50], while the WAL is governed by SOI. Thus the change in the MR characteristics upon changing the thickness of the STO layer reflects a significant modulation of the SOI in the heterostructures[24,25,42,51].

To quantify the modulation of SOI upon the change of STO layer thickness, the model developed by Iordanskii, Lyanda-Geller, and Pikus (ILP), which considers the **k**-dependent SOI, was adopted to analyze the magnetotransport data[52,53]. This model involves two spin-splitting energy terms due to different **k**-dependent spin-precession vectors, i.e., one is the linear SOI term and the other is the cubic SOI term. The full equation of the ILP model can be written as follows:[52]

$$\Delta\sigma(B) - \Delta\sigma(0) =$$

$$-\frac{e^2}{4\pi^2\hbar}\left[\frac{1}{a_0} + \frac{2a_0 + 1 + \frac{B_{so1}}{B} + \frac{B_{so3}}{B}}{a_1\left(a_0 + \frac{B_{so1}}{B} + \frac{B_{so3}}{B}\right) - 2\frac{B_{so1}}{B}}\right.$$

$$-\sum_{n=1}^{\infty}\left\{\frac{3}{n} - \frac{3a_n^2 + 2a_n\left(\frac{B_{so1}}{B} + \frac{B_{so3}}{B}\right) - 1 - 2(2n+1)\frac{B_{so1}}{B}}{\left(a_n + \frac{B_{so1}}{B} + \frac{B_{so3}}{B}\right)a_{n-1}a_{n+1} - 2\frac{B_{so1}}{B}[(2n+1)a_n - 1]}\right\}$$

$$+ \Psi\left(0.5 + \frac{B_\phi}{B}\right)\right] - \frac{e^2}{2\pi^2\hbar}\left[-0.5\ln\frac{B_\phi}{B} + \ln\frac{B_\phi + B_{so1} + B_{so3}}{B}\right.$$

$$+ 0.5\ln\frac{B_\phi + 2B_{so1} + 2B_{so3}}{B}\right],$$

$$(1)$$

where

$$a_n = n + \frac{1}{2} + \frac{B_\phi}{B} + \frac{B_{so1}}{B} + \frac{B_{so3}}{B}. \qquad (2)$$

In the equation, $B_{so1}$, $B_{so3}$, and $B_\phi$ refer to the characteristic effective magnetic fields for the linear SOI, the cubic SOI, and the phase coherence, respectively, and $\Psi$ is the digamma function. Here the diffusion limit of this model is satisfied as the applied magnetic field is much smaller than the $B_e$, the characteristic field of the elastic scattering, which is of the order of 100 T because of the short mean free path in our heterostructures. To account for the classical Lorentz force, which manifests itself as quadratic dependence in field, the resistivity data are fitted directly using

$$R_{xx}(B) = \frac{\gamma}{((\Delta\sigma(B) - \Delta\sigma(0)) + \delta)} + \alpha B^2, \qquad (3)$$

where $\alpha$ is the coefficient for the MR due to the Lorentz force, $\gamma$ is a constant, $\ln(2)/\pi$, due to the Van der Pauw method used for the transport measurements, and $\delta$ is the field-independent component of conductivity of the samples. The data were symmetrized to avoid any artifact due to the choice of magnetic field direction. The analysis of the fitting accuracy can be found in Supplementary Fig. 6. Compared to the fitting with only a cubic SOI term, the fitting considering both linear and cubic SOI terms gives a more accurate result.

Figure 5b shows the fitting result of the magnetoconductance data using the ILP model, which are obtained after removing the classical Lorentz components, and the derived fitting parameters are shown in Fig. 5c. One main finding of this work is that $B_{so3}$ monotonically increases from 1 to 3.75 T as the thickness of the STO layer decreases from 60 uc to 8 uc in the LAO//STO/LAO heterostructures. The sample-dependent $B_\phi$ fluctuates in the range of 1–2 T, which is smaller than the characteristic $B_{so3}$ and thus in

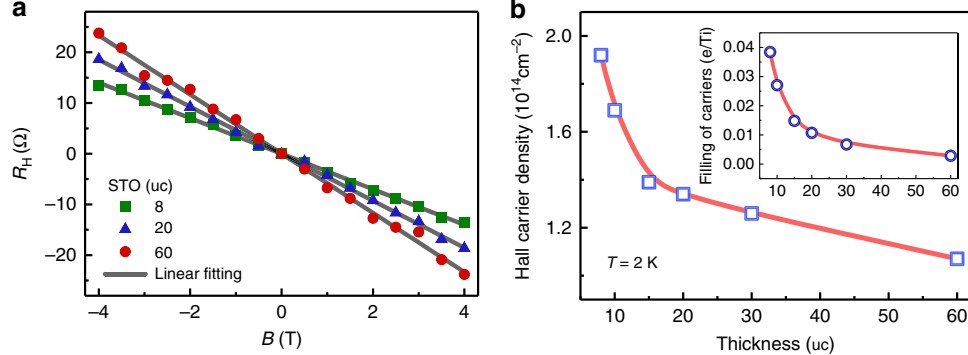

**Fig. 3** Carrier density estimation of the LAO//STO/LAO heterostructures. **a** Hall effect data measured at 2 K for the LAO//STO/LAO heterostructure with 8, 20, and 60 uc STO layers along with the linear fitting. **b** STO thickness-dependent Hall carrier density, from which the filling of electrons per Ti is calculated (inset). The solid lines are guides to the eyes

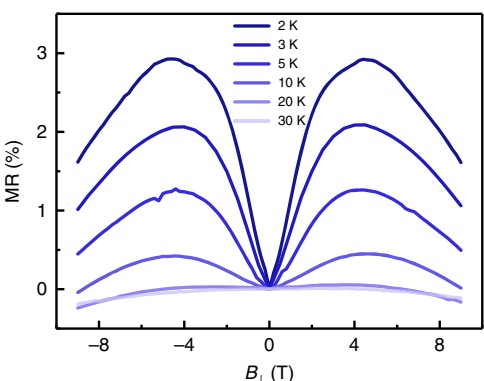

**Fig. 4** Temperature-dependent magnetoresistance. Magnetoresistance measured on the LAO//STO/LAO (10//30/10 uc) heterostructure at various temperatures

line with the WAL mechanism[50]. More significantly, $B_{so1}$ emerges only when the thickness of the STO layer is below 20 uc; it monotonically increases to 2.13 T in the thinnest heterostructure with 8 uc STO. The temperature-dependent characteristic fields have been extracted too, as shown in Supplementary Fig. 7. Our data indicate that the prerequisite of the emergence of the linear Rashba SOI is a high doping level of 3d electrons in thin STO layers. On the other hand, the cubic SOI was found to dominate in the heterostructures with STO thickness above 20 uc, similar to the recent report by Nakamura et al.[41]. The temperature-dependent resistance for all the samples are shown in Supplementary Fig. 8, where an upturn behavior of resistance was observed at low temperatures. Furthermore, the X-ray linear dichroism (XLD) experiments has been performed to confirm that the lowest orbital in the LAO//STO/LAO structure is $d_{xz/yz}$ and the electron occupation changes upon the thickness of the STO layer (Supplementary Note 5 and Fig. 9). The XLD result strongly supports the scenario that the Rashba effect is tuned in the LAO//STO/LAO structures by carrier filling of the Ti 3d orbitals.

Accordingly, the thickness dependence of spin diffusion lengths due to the linear and cubic SOI terms was calculated using the equation $l_{so} = \sqrt{\hbar/4eB_{so}}$[50]. As described in Supplementary Note 6, we calculated the linear Rashba coefficient $\alpha_{\text{linear}}$ and the spin splitting energy $\Delta_{\text{cubic}}$ from the cubic SOI term for the heterostructures. The linear Rashba coefficient increases from 0.015 to 0.043 eVÅ upon a decrease in the STO layer thickness from 15 uc to 8 uc, which is in the same range as the results of

recent reports[25,40]. When the thickness of the STO layer decreases from 60 uc to 20 uc, the spin splitting energy due to the cubic SOI term increases from 2.31 to 2.87 meV, and concurrently the electron number per Ti increases from 0.003 to 0.01. The measured spin splitting energy is consistent with the first-principles calculation result (Fig. 1d). To elucidate the relationship between the SOI transition and the electron doping level of Ti, we plotted the above extracted parameters as functions of the filling electrons per Ti determined from the Hall measurement, as presented in Fig. 5d. It can be seen that the transition between the linear and cubic Rashba SOI occurs at a doping level of approximately 0.01–0.015 e/Ti, which is consistent with the value predicted by our first-principles calculation result.

In conclusion, we demonstrated a class of oxide heterostructures with interface-based broken inversion symmetry, in which the Rashba effect can be tuned by changing the thickness of the intermediate layer with itinerant 3d electrons. The lifted orbital degeneracy and carriers' doping modulated by the STO layer thickness in the LAO//STO/LAO heterostructures enabled us to identify the transition between the cubic and linear Rashba effects. This work offers an alternative heterostructure-based route to manipulating the SOI, complimentary to the reports on electric field effect. Furthermore, our study also reveals the unambiguous role of the linear Rashba SOI in thin heterostructures, while the previous reports were focused on cubic SOI at oxide interfaces. The coexistence of the linear and cubic Rashba effects in such oxide heterostructures with 3d electrons will stimulate further theoretical and experimental studies. Such asymmetric heterostructures represent an alternative platform for exploring SOI and other exotic physics in artificial quantum materials.

## Methods

**DFT-based tight-binding calculations**. To construct a realistic tight-binding model for STO-based heterostructure and also avoid adjustable parameters, we performed a projection[54] of Wien2K[55] DFT results for bulk STO onto maximally localized Wannier orbitals, which exactly reproduced the DFT-calculated band structure[40,56]. The tight-binding Hamiltonian is described by $H_o + H_\xi + H_\gamma$ in the $t_{2g}(xy, yz, xz)$ basis. The $H_o$ term contains local energy terms $\varepsilon$ and hopping terms $t$. In bulk STO with cubic symmetry $\varepsilon_{xy} = \varepsilon_{yz/xz}$, which is defined with respect to Fermi energy level, the compressive strain from the LAO substrate increases the $xy$ orbital energy in the order of 10 meV. Without loss of generality, we set $\varepsilon_{xy} - \varepsilon_{xy/xz}$ = 20 meV. There are three hopping terms: the large hopping term of $t_1 = 0.277$ eV arises from the large $xy$ intraorbital hopping integral along the $x$ and $y$ directions; $t_2 = 0.031$ eV and $t_3 = 0.076$ eV indicate a much smaller hopping integral along the $z$ and (1,1,0) directions of the $xy$ orbital, respectively. The $H_\xi$ term includes atomic SOI, whose strength is 19.2 meV, as estimated from the DFT calculated orbital splitting at $\Gamma$ point. The last term $H_\gamma = <xy|H|yz/xz>$ is an antisymmetric hopping between $xy$ and $yz/xz$ orbitals along the $x/y$ direction. It describes the asymmetry due to the built-in electric field, which results in the Rashba spin splitting. In the bulk STO, $H_\gamma$ vanishes due to inversion symmetry, while at the LAO/STO interface

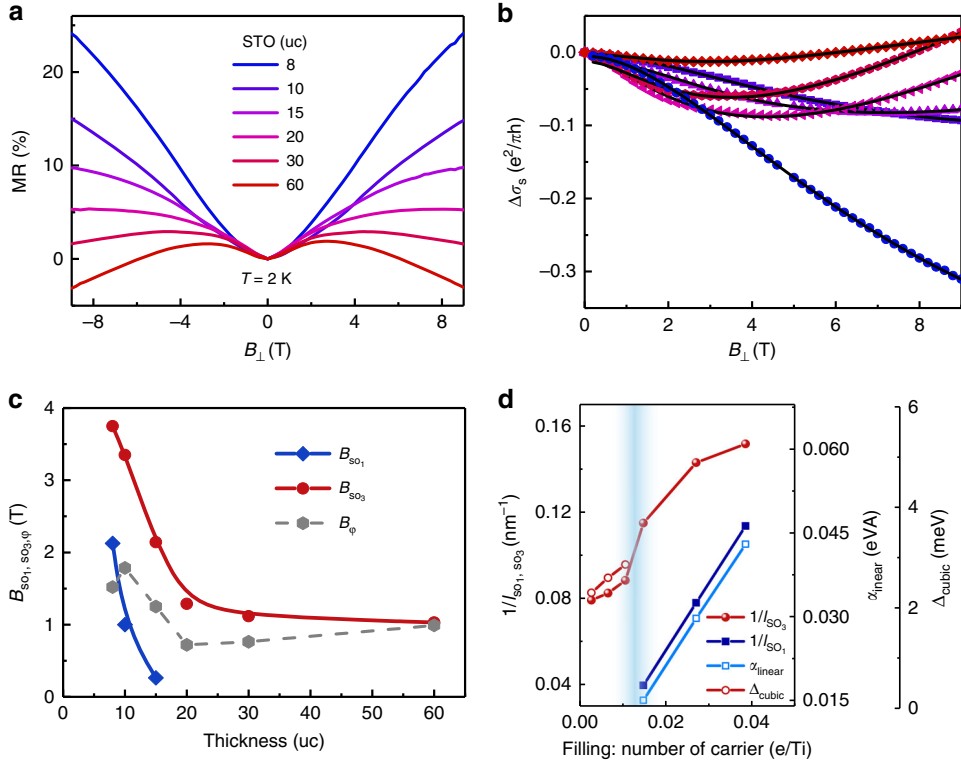

**Fig. 5** Spin splitting modulated by carrier-filling. **a** Evolution of magnetoresistance at 2 K for the LAO//STO/LAO oxide heterostructures with various STO thicknesses. **b** Fitting (lines) of the magnetoconductance data (symbols) to the Iordanskii, Lyanda-Geller, and Pikus model. **c** Thickness-dependent fitting parameters: $B_{so1}$ and $B_{so3}$ are the characteristic fields of the linear and cubic Rashba terms, respectively, and $B_{\varphi}$ is the characteristic field for the phase coherence. **d** Inverse spin relaxation lengths, linear Rashba coefficient $\alpha_{linear}$, and spin splitting energy $\Delta_{cubic}$ as a function of the filling carriers per Ti in the heterostructures

$H_{\gamma}$ is 20 meV. It is noted that the Rashba transition as well as the corresponding critical carrier concentration mainly depend on $H_{\xi}$ and the $t_{2g}$ band structure character of STO. It is confirmed that a change of hopping terms $t$ or crystal field splitting will not influence our conclusion.

**Sample preparation and characterization.** To ensure the $AlO_2$ termination of the LAO substrate, all the substrates used were treated and confirmed by atomic force microscopic mapping before growth. The LAO//STO/LAO heterostructures were grown in a PLD system, and high-pressure RHEED was used to monitor the growth quality and to control the thickness of the growing films. Both LAO and STO layers were grown at 800 °C under a pressure of $1 \times 10^{-6}$ mbar of $O_2$ and with the laser energy set at 1 J cm$^{-2}$. The laser pulse was set at a 1-Hz repetition rate. After deposition, the samples were cooled to room temperature at 5 °C min$^{-1}$. With the same growth conditions, a series of heterostructures with controlled STO thicknesses were prepared. Van der Pauw method was used to perform the transport measurements with Ti/Au as the electrodes. All the transport measurements were conducted in a physical property measurement system (PPMS, Quantum Design). STEM was performed using a JEOL ARM200F operating at 200 kV and equipped with ASCOR probe corrector and Oxford XX-Max 100TLE X-ray detector.

## Data availability

The data that support the findings of this study are available from the corresponding author upon reasonable request.

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

## Acknowledgements

L.L. and Z. Z. gratefully acknowledge financial support from the National Key R&D Program of China (2017YFA0303602), 3315 Program of Ningbo, and the National Nature Science Foundation of China (11774360). Calculations were performed at the Supercomputing Center of Ningbo Institute of Materials Technology and Engineering. W. L. and J. C. acknowledge the financial support from the Singapore National Research Foundation under CRP Award No. NRF-CRP10-2012-02 and Singapore Ministry of Education MOE2018-T2-2-043, AMEIRG18-0022, A*STAR IAF-ICP 11801E0036 and MOE Tier 1- FY2018–P23. C.J.L. acknowledges the financial support from the Lee Kuan Yew Postdoctoral Fellowship through the Singapore Ministry of Education Academic Research Fund Tier 1 (Grant No. R-284-000-158-114). F. D. and A. M. acknowledge valuable support from KAUST Supercomputing team and some of the fitting calculations were performed on the Phoenix High Performance Computing facility at the American University of the Middle East, Kuwait. W.P. acknowledges partial support of the Tan Chin Tuan Exchange Fellowship in Engineering and Merlion project .The authors would like to acknowledge the Singapore Synchrotron Light Source (SSLS) for providing the facility necessary for conducting the research. The SSLS Laboratory is a National Research Infrastructure under the Singapore National Research Foundation.

## Author contributions

W.L., T.W., and Z.Z. conceived and designed the experiments. W.L. performed the samples' fabrication, transport measurements, and data analysis. L.L. and Z.Z. performed the DFT-based tight-binding calculation. F.D. and A.M. contributed to analyze the magnetotransport data. C.L. and S.J.P. performed STEM experiments. X.Y. performed the XLD experiment. H.R., W.P., B.Z., and Y.L. helped on the structure analyses. W.S.L. and S.W. helped on the samples' fabrication. W.L., T.W., L.L., and Z.Z. wrote the manuscript and all authors contributed to its final version. T.W., J.C., Z.Z., and A.M. supervised and guided the project.

## Additional information

**Competing interests:** The authors declare no competing interests.

