## [Peer Review File · Nature Communications]

Reviewers' comments:

Reviewer #1 (Remarks to the Author):

It is a very interesting article dealing about magneto-transport - Rashba SOI in STO/LAO related system.

They clearly evidence that they can tune / modulate the Rashba SOI using a different geometry, Id est, LAO/STO/LAO. For this system, the STO orbital ordering is inverted (yz/xz lower than xy) as compared to the traditional LAO/STO, resulting in different linear vs cubic Rashba SOI possibility. Furthermore they clearly show that the effect can be tuned by the STO thickness and band filling. Modeling, microscopy and magneto-transport are convincing and well explained.

Reviewer #2 (Remarks to the Author):

In this manuscript, the authors report on the observation of a transition between two different regimes of Rashba spin-orbit interaction in an asymmetric SrTiO₃-based oxide heterostructures. The overall context of the study is quite timely and I find the experimental signatures reported here interesting for the community. However, at this stage I am not convinced that the measurements and the analysis are sufficient to support the conclusions of the paper. My main criticism is that the claims entirely rely on the fitting of magneto-resistance curves through a complex formula (ILP model) that, in my opinion, includes too many fitting parameters to be relevant. In addition, several critical assumptions in their approach are not justified. I cannot give my recommendation to publish this article in Nature Communications, unless the authors find a way to strengthen their analysis, and provide additional experimental signatures of the cubic to linear Rashba transition.

- In the context of LAO/STO interfaces, magneto-conductance curves are often fitted by Maekawa-Fukuyama formula which is also derived within a weak-localisation framework (for instance in ref 25 and 51). How does that compare with the ILP model used by the authors ?

In ref 25, the authors also include a Zeeman splitting term, which is absent in the ILP model.

- In ref 51, the authors introduce also a "Kohler" term that account for classical magneto-resistance. This should correspond to the alpha term in front of the B² term in the formula given line 238. However, the authors have used an approximation and not the full magnetic field dependence that includes a saturation of classical magneto-resistance at high magnetic field. The MR curves of thin STO layers (8 uc, 10 uc) could be very well fitted by adding a full "Kohler" term and a single BSO term making the discussion on the cubic to linear Rashba transition irrelevant. This term could be negligible but this must be justified.

- Related to the previous comments, I recommend the authors to clarify the meaning of the gamma, delta and alpha coefficients in the expression of the resistance (line 238). I also would like to see the value of these fitting parameters for the different thicknesses of STO.

- One way to disentangle the BSO terms from the Bphi one and give consistency to the analysis is to look at the temperature dependence of the magneto-resistance in the temperature range of validity of WL theory (<10K). The Bphi term should display a power law type of variation with temperature with a characteristic exponent that depends on the scattering mechanism that limits phase coherence (e-e, e-phonon). On the other hand, BSO terms, which only depend on the band structure, should not be affected by the temperature.

- Hall effect in LAO/STO interfaces is known to be non-linear at high doping level because of multi-band transport. According to fig 1, the cubic to linear Rashba transition takes place between two

bands (red and blue) of different masses. I would therefore expect very different mobilities in the two bands that should result in a transition between a linear Hall effect and a non-linear one at doping around $0.01 e/Ti$.

- In their calculation, the authors assume a perfect crystal structure, which is in contradiction with the structural analysis. This aspect is really critical as the main claim (transition between two regime of Rashba SO) relies on an apparent agreement between theory and experiment. I can hardly imagine that the transition at $0.01e/Ti$ is sufficiently robust to any kind of imperfection in the structure.

- The authors assume that electrons are homogeneously distributed in the STO. I would expect that for the highest thicknesses of STO, one should recover the same situation than in LAO/STO heterostructures, with electrons confined at the interfaces. In this case, the low doping regime corresponding to the transition at $0.01e/Ti$ could be a pure artefact.

- The resistance vs temperature curves should be shown for the different samples. Do they exhibit the characteristic upturn in the resistance expected for WL ?

- The authors should clearly indicate what is the exact formula used in the fit of magneto-resistance.

Reviewer #3 (Remarks to the Author):

- Key results: The paper of Lin et al. reports the study of the transport properties of $LaAlO_3/SrTiO_3/LaAlO_3$ heterostructures formed by two non-equivalent interfaces. They propose this system as a good candidate to study the spin-orbit Rashba effect, which is a really interesting property for the spintronic community.

One of the main results given by this work is that, by varying the $SrTiO_3$ layer thickness, the authors achieved to tune the carrier density and observed experimentally a transition from a cubic to linear (more precisely cubic+linear, if I understood correctly) Rashba effect, the linear Rashba effect being only predicted theoretically until now.

- Validity: The manuscript is globally well written. Some precisions seem however necessary to judge about the complete validity of the paper.

- Originality and significance: The observation of the linear Rashba effect will undoubtedly motivate further study to get a better understanding of this system and to get a better control of this interaction. It would thus be interesting to submit it for discussion. It is however not certain that this paper will get an audience out of the spintronic and oxide community.

- Data & methodology: The magneto-transport measurements are associated with some analysis of the atomic structure performed by STEM and some support from first-principles calculations.

For the DFT-based tight binding calculations, I would suggest to add more information about the parameters which were used and the methodology. In particular, it will be easier for the reader to have everything in hand than to collect all the information from previous references.

What is really from "first principles" and which parameters (energy splitting, atomic spin-orbit interaction, asymmetry strength coefficient) are chosen to fit with the experience? In Ref. 40, some parameters are obtained from DFT calculations, but performed on systems with different geometries and in-plane lattice parameters. Was it modified for the current paper? L. 142-143, the authors say that the results are supported by DFT calculations and refer to the supplementary information. I did not see any detail about the DFT calculations (which code was used, which Exc functional...? (Wien2k + GGA?)). How was the value given in Fig. S4d (purple square) obtained?

With a slab geometry to simulate the effect of the internal electric field (with a STO thickness of 3-4 u.c.), and an in-plane lattice parameter equal to the experimental or calculated one? How are defined the values represented by the dashed lines in the same figures (also calculations?).

About the chemical analysis, it is said that the two interfaces show clear cation intermixing, but that they remain non-equivalent. Was such analysis performed for different STO layer thicknesses? Or only for the 20 u.c.? From what is said, the out-of-plane STO layer lattice parameter is lower than expected mainly due to electrostrictive effect. Such effect should be linked with the asymmetry of the interfaces, and thus it should depend on the atomic structure at these interfaces. Did the authors notice any difference for the low and high layer thicknesses (that is, below and above 20 u.c., the thickness at which the transition occurs)?

The "rough" approximation which consists in averaging the number of electrons per Ti should be maybe discussed a little. The localization of the electrons near the interface is important as it depends on the internal electric field, which will itself have some impact on the Rashba splitting. Moreover, such approximation is quite contradictory with the fact that all the study relies on having two non-equivalent interfaces.

Is it possible to have some information on the method used to fit the parameters for Fig. S6? Above 20 u.c. the red and black points are superimposed, right? Why is the error increasing (of a factor 3) above 20 u.c.? This increase of the error is higher than the difference of errors between the fits with and without the linear contribution.

In the supplementary paragraph S4, how are the effective mass determined?

Also, I have two questions for my understanding of the results:

1) If the carrier in STO are mainly here due to the polarity of LAO, can the authors explain a little more why the carrier density varies as a function of the STO thickness?

2) Why do the authors finally show some evidence of the linear Rashba effects, while other did not? Is it linked with the high carrier density? The special geometry of the system? From the bandstructure, I can understand why the cubic Rashba is seen first, and the linear appears only after, but if we forget the band energy ordering, the linear Rashba should be also present for the standard LAO/STO interface (?). I did not get the hypothesis that the dxy orbital being localized at the interface, the corresponding electrons would not participate to the charge transport.

- Conclusions: Regarding my previous comments, most of the conclusion are certainly reliable, but some of them would greatly benefit of more details (in particular about the methods).

- References: ok

- Clarity and context: In the abstract, I would suggest to change "transition between cubic and linear" by "transition between cubic and linear+cubic".

- Particular part of the manuscript that you feel is outside the scope of your expertise: I am not expert in transport measurement and the full meaning of the ILP equations and their use is quite new for me. Sorry if some of my questions were "naive" in this sense.

Response to the referees' comments

(The referee's comments are quoted and our point-to-point responses in blue. Changes made in the revised manuscript are marked in blue.)

Reviewer #1 (Remarks to the Author):

It is a very interesting article dealing about magneto-transport - Rashba SOI in STO/LAO related system. They clearly evidence that they can tune / modulate the Rashba SOI using a different geometry, Id est, LAO/STO/LAO. For this system, the STO orbital ordering is inverted (yz/xz lower than xy) as compared to the traditional LAO/STO, resulting in different linear vs cubic rashba SOI possibility. Furthermore they clearly show that the effect can be tune by the STO thickness and band filling.

Modeling, microscopy and magneto-transport are convincing and well explained.

Response: We thank the positive evaluation of our work from the reviewer.

Reviewer #2 (Remarks to the Author):

In this manuscript, the authors report on the observation of a transition between two different regimes of Rashba spin-orbit interaction in an asymmetric SrTiO₃-based oxide heterostructures. The overall context of the study is quite timely and I find the experimental signatures reported here interesting for the community.

Response: We thank the reviewer for his/her recognition of the value of our study.

However, at this stage I am not convinced that the measurements and the analysis are sufficient to support the conclusions of the paper. My main criticism is that the claims entirely rely on the fitting of magneto-resistance curves through a complex formula (ILP model) that, in my opinion, includes too many fitting parameters to be relevant. In addition, several critical assumptions in their approach are not justified. I cannot give my recommendation to publish this article in Nature Communications, unless the authors find a way to strengthen their analysis, and provide additional experimental signatures of the cubic to linear Rashba transition.

Response: We thank the reviewer for these critical comments. We agree with the reviewer that the fitting to the magnetotransport data is quite complex, but it is still a reliable method and has been utilized to investigate the SOC in various systems, such as 2DEG in semiconductor quantum well, two dimensional materials and oxide heterostructures [G. Bergmann, Phys. Rep. 107, 1-58(1983), P. A. Lee, and T. V.

Ramakrishnan Rev. Mod. Phys 57, 287-337(1985), H. Yuan *et al*, Nat. Phys. 9, 563–569 (2013), H. Nakamura, T. Koga, and T. Kimura, Phys. Rev. Lett. 108, 206601(2012), R. Moriya *et al*, Phys. Rev. Lett. 113, 086601(2014), T. Koga *et al*, Phys. Rev. Lett. 89, 046801(2002)]. Nevertheless, to address the concern of the referee, we improved the manuscript on the following three aspects:

First, we have provided more details about the magnetoresistance fitting and the RT data to further validate the method used in this manuscript.

Second, we performed X-ray linear dichroism (XLD) experiments on the LAO//STO/LAO structures with 8 uc and 30 uc STO, as shown in Fig.R1. The XLD data confirm that the $d_{xz/yz}$ orbitals are the first available states in the LAO//STO/LAO structure as predicted, while the d_{xy} is the lowest state in the conventional LAO/STO structures. And as the STO thickness decreases, more electrons occupy the $d_{xz/yz}$ orbitals, which support the modulation effect of the STO thickness. The details of the XLD experiment are as follow:

Linearly polarized X-ray absorption spectroscopy (XAS) experiments were carried out by adjusting the incident angle of the X-ray beam. As shown in Fig. R1a, the in-plane (IP) component is obtained by the normal incident X-ray, while the out-of-plane (OP) one by the grazing incident X-ray. All spectra were acquired by recording the total electron yield (TEY) at Ti $L_{2,3}$ -edge. These linearly polarized X-rays will excite the electrons from the Ti 2p core level of 2p $1/2$ and 2p $3/2$ states to the unoccupied d orbital, and thus the intensity of the XAS (I_{IP} and I_{OP}) reflects population of the empty states. Moreover, the absorption of the X-ray shows strong dependence on the photon polarization with respect to the lattice direction, i.e. IP and OP components will excite more electrons to the d_{xy} (dx^2-y^2) and $d_{xz/yz}$ (dz^2), respectively, for the electrons from the Ti 2p core level. Therefore, the sign of XLD data, ($I_{IP} - I_{OP}$), indicates the state of the electron occupation of the d orbitals in the LAO//STO/LAO structures. As shown in Fig. R1b and c, the XLD sign at the four main peaks are positive, indicating more IP orbitals (i.e. d_{xy}) are available [D. Pesquera *et al*, Phys. Rev. Lett. 113, 156802(2014)]. This means that the $d_{xz/yz}$ orbital is occupied first, which is consistent with the calculation. Such a feature with different electrons populations in different orbitals is called orbital polarization, whose strength can be characterized as $2(I_{IP} - I_{OP}) / (I_{IP} + I_{OP})$. The strengths for the structures with 30 uc STO and 8 uc STO at the t_{2g} main peak of L_3 edge are 3.50% and 4.94%, respectively, meaning more electrons occupy the $d_{xz/yz}$ orbitals at the structure with 8 uc STO. This is consistent with our Hall measurement data that the Fermi level increases as the STO thickness decreases.

Third, as presented in the manuscript, the density-functional-theory (DFT)-based tight-binding (TB) calculation supports the interpretation of the magnetotransport measurements. In the revised manuscript, we provide more details on the calculation, as suggested by the Reviewer 3.

Fig. R1 | X-ray linear dichroism (XLD) measurements of LAO//STO/LAO at Ti L_{2,3}-edge. (a) Schematic experimental set-up for the linearly polarized X-ray absorption spectroscopy (XAS) at Ti L_{2,3}-edge with the total electron yield (TEY) detection mode at room temperature. In this measurement configuration, the polarization direction of the linearly polarized X-rays was achieved by modulating the X-ray incidence angle. Here, the X-ray with normal and grazing (30°) incidence correspond to the in-plane (E // IP) and majority out-of-plane (E // OP) polarized components, respectively. (b) and (c) show the XAS data for the LAO//STO/LAO samples with 30 uc and 8 uc STO, respectively. The XLD signal (green curves) are obtained from (I_{IP}-I_{OP}).

To give more support of the proposed scenario in the designed LAO//STO/LAO structure, we have added the XLD experiment in the revised supplementary materials (supplementary information 5 and Fig. S9).

- In the context of LAO/STO interfaces, magneto-conductance curves are often fitted by Maekawa-Fukuyama formula which is also derived within a weak-localisation framework (for instance in ref 25 and 51). How does that compare with the ILP model used by the authors?

In ref 25, the authors also include a Zeeman splitting term, which is absent in the ILP model.

Response: Hikami-Larkin-Nagaoka (HLN) model is the pioneering work on investigating the quantum correction of the magnetoconductance, i.e. weak (anti-)localization [S. Hikami, A. I. Larkin, and Y. Nagaoka, Prog. Theor. Phys. 63, 707 (1980)]. Maekawa-Fukuyama model was developed later with the further consideration of the Zeeman effect, and it is suitable for describing spin orbit interaction (SOC) with moderate or strong strength [S. Maekawa and H. J. Fukuyama, Phys. Soc. Jpn 50, 2516–2524 (1981)]. Generally, the Zeeman effect will be suppressed with the perpendicular magnetic field if a strong SOC is present in the system. Therefore, the Maekawa-Fukuyama formula with the Zeeman term

neglected, which is equivalent to the HLN model, can still be used for the magnetoconductance fitting (Ref. 51). However, both models do not consider the k-dependent SOC, which leads to the development of the Iordanskii, Lyanda-Geller, and Pikus (ILP) model [S. V. Iordanskii, Y. B. Lyanda-Geller, and G. E. Pikus, JETP Lett. 60, 206 (1994)]. For the ILP model, if the linear Rashba term is removed, the model can be reduced to the HLN formula. Until now, the ILP model has been successful used to investigate the SOC in various systems, such as WSe₂, SrTiO₃ surface and Ge/SiGe quantum Well [H. Yuan *et al*, Nature Phys. 9, 563–569 (2013), H. Nakamura, T. Koga, and T. Kimura, Phys. Rev. Lett. 108, 206601(2012), R. Moriya *et al*, Phys. Rev. Lett. 113, 086601(2014)].

To specify the reason we chose the ILP model, we add a sentence “which considers the k-dependent SOI” in second paragraph of page 11.

- In ref 51, the authors introduce also a "Kohler" term that account for classical magneto-resistance. This should correspond to the alpha term in front of the B² term in the formula given line 238. However, the authors have used an approximation and not the full magnetic field dependence that includes a saturation of classical magneto-resistance at high magnetic field. The MR curves of thin STO layers (8 uc, 10 uc) could be very well fitted by adding a full "Kohler" term and a single BSO3 term making the discussion on the cubic to linear Rashba transition irrelevant. This term could be negligible but this must be justified.

Response: Thanks for the comment. We agree with the reviewer that if the magnetic field is strong enough, the B² term might be not applicable, which may influence our fitting model for the magnetoconductance data. However, to disqualify the B² term, the magnetic field should satisfy that $\mu H > 1$, wherer μ is mobility. Therefore, for the maximum magnetic field (9 T) applied in MR measurement, this will require $\mu > 1.1 \cdot 10^3 \text{ cm}^2 \text{ V}^{-1} \text{ s}^{-1}$, which is far larger than the mobility of the carriers in the LAO//STO/LAO structures. Second, we aware that Kohler’s rule is the relationship that governs how the applied transverse magnetic field modifies the resistance of a materials, and can be summarized as $\frac{\Delta\rho}{\rho_0} = f\left(\frac{H}{\rho_0}\right)$, where f is a function that is related to electronic properties of the sample [P. Lorrain, D.R. Corson, and F. Lorrain, Electromagnetic fields and waves, 3rd ed., W.H. Freeman and company (1988)]. The B² term used here is its simple application for the Lorentz force [Sirisathitkul, C., et al. J. Sci. Technol., 24, 305-310 (2002)]. To have the conductivity expression in ref. 51, there are two assumptions required: one is that the maximum applied magnetic field is strong enough (around the saturation, unsupported by the argument above), which will result in the resistivity expression as $\frac{\Delta\rho}{\rho_0} = \frac{A(H/\rho_0)^2}{1+C(H/\rho_0)^2}$ [Feng Duan, and Jin Guojun. Introduction to Condensed Matter Physics: Volume 1. World Scientific Publishing Company (2005)]; Second, to convert the resistivity expression to the conductivity one as

shown in ref. 51 ($\frac{\Delta\sigma}{\sigma_0} = \frac{AH^2}{1+CH^2}$), the contribution from the Kohler term should be assumed to be very small, which is not made in our study.

Therefore, we do not think that it is reasonable to include the full magnetic field dependent Kohler term in the fitting model, which will also complicate the fitting process as more fitting parameters are involved (the model in ref 51).

- Related to the previous comments, I recommend the authors to clarify the meaning of the gamma, delta and alpha coefficients in the expression of the resistance (line 238). I also would like to see the value of these fitting parameters for the different thicknesses of STO.

Response: Thanks very much for the suggestion. Among these three parameters, γ is the reciprocal Van der Pauw constant due to the method we used here, which is a constant, $\ln(2)/\pi$, for all the fitting procedures. α is the coefficient for the magnetoresistance due to the Lorentz force and δ is the field independent component of conductivity of the samples. We have updated the meanings for the all these parameters in the revised manuscript, page 12. The fitted parameters of δ and α are listed in the table below.

Table R1 | Fitting parameters, δ and α , as function of the thickness of STO.

STO thickness (uc)	δ ($e^2/(\pi h)$)	α (Ω/T^2)
8	1.28	3.77
10	1.19	8.97
15	1.42	5.99
20	3.11	2.30
30	2.53	2.37
60	1.41	4.43

- One way to disentangle the BSO terms from the Bphi one and give consistency to the analysis is to look at the temperature dependence of the magneto-resistance in the temperature range of validity of WL theory (<10K). The Bphi term should display a power law type of variation with temperature with a characteristic exponent that depends on the scattering mechanism that limits phase coherence (e-e, e-phonon). On the other hand, BSO terms, which only depend on the band structure, should not be affected by the temperature.

Response: Thanks very much for the comment. We agree with the reviewer that the temperature dependent behaviour of the extracted fitting parameter can help disentangle $B_{\text{so1/so3}}$ and B_{φ} . Figure R2 shows the temperature dependent fitting parameters B_{so1} , B_{so3} and B_{φ} for the structures with 8 uc and 30 uc STO, respectively. For both structures, the B_{φ} increases as the temperature increases, which is consistent with the behavior of the phase coherence property of the conducting carriers. The SOI strength, characterized by B_{so1} and B_{so3} , of both samples are not very sensitive to the temperature, except the ones at 5 K of the sample with 8 uc STO. The deviation might be due to the influence of the thermal fluctuation to the quantum correction to the conductance (The coexistence of the cubic and linear Rashba terms might be more sensitive to the thermal fluctuation). We have added the temperature dependent effective fields for structures with 8 uc and 30 uc STO in the revised supplementary materials as Fig. S7.

Figure R2 | Temperature dependent fitted effective fields for the LAO//STO/LAO with 8 uc (a) and 30 uc (b) STO.

- Hall effect in LAO/STO interfaces is known to be non-linear at high doping level because of multi-band transport. According to fig 1, the cubic to linear Rashba transition takes place between two bands (red and blue) of different masses. I would therefore expect very different mobilities in the two bands that should result in a transition between a linear Hall effect and a non-linear one at doping around $0.01 e/\text{Ti}$.

Response: Thanks for the comment. We agree with the reviewer that when a system have carriers from two bands with very different mobilities, it is highly possible to result in a non-linear Hall effect. In the conventional LAO/STO interface, the non-linear Hall effect was observed with the carriers from d_{xy} and $d_{xz/yz}$ (or light band and heavy band) coexisting, normally achieved by the gating effect [A.E.M. Smink *et*

al, Phys. Rev. Lett. 118, 106401 (2017), J. Biscaras *et al*, Phys. Rev. Lett. 108, 247004 (2012), M. Ben Shalom *et al*, Phys. Rev. Lett. 105, 206401 (2010), A. Joshua *et al*, Nat. Commun. 3, 1129 (2012)]. In these cases, the extracted mobilities of the carriers from both bands differ by three order of magnitudes. Such a huge difference of the mobilities are not due to the different effective masses of the orbitals ($1m_0$ vs several m_0), but the different extensions of this two type of carriers, i.e. one is confined in the interface and the other can extend into the bulk STO. As the electrostatic gating can effectively modulate the extension of the carrier in the conventional LAO/STO heterostructures, most of the non-linear Hall effect were discussed in the context of such experiments [C. Bell *et al*, Phys. Rev. Lett. 103, 226802 (2009), Z. Chen *et al*, Nano Lett. 16, 6130-6136 (2016)]. In contrast, in the LAO//STO/LAO asymmetric structures, the redistribution of carriers from the two $d_{xz/yz}$ orbitals is not that significant when the Rashba transition occurs. Therefore, the mobilities of the carriers from these two orbitals are expected to exhibit only modest change due to their different effective masses. As a result, the non-linear Hall effect was not observed in our structures when the Rasha transition occurs.

- In their calculation, the authors assume a perfect crystal structure, which is in contradiction with the structural analysis. This aspect is really critical as the main claim (transition between two regime of Rashba SO) relies on an apparent agreement between theory and experiment. I can hardly imagine that the transition at $0.01e/Ti$ is sufficiently robust to any kind of imperfection in the structure.

Response: Thanks very much for the comment. From the TEM results in the manuscript, the interfaces at the LAO//STO/LAO structure are not perfect. However, we do not think that the imperfect interfaces will significantly alter the critical carrier concentration where the Rashba transition occurs. The transition at $0.01e/Ti$ is determined by the electronic band structure of the STO layer, i.e. the energy splitting of $d_{xz/yz}$ at the Γ point depends on the atomic SOC strength of the Ti atom, which is an intrinsic properties of the STO crystal structure, and less sensitive to the imperfection of the interfaces unless the imperfect interfaces could significantly alter the electronic band structure of the entire STO layers. The influence of the imperfect interfaces may be on the scattering event of the conducting carriers, their mobility, lifetime of the quasi-particles, and so on. From our mangetotransport measurements, the transition between the two Rashba regimes occurs at around 0.01-0.015 e/Ti at the structure with STO thickness between 15 and 20 uc. The deviation from the calculated value, $0.01e/Ti$, might be due to the average filling number obtained from the Hall measurements.

To clarify this, we added one sentence in the revised manuscript “We should note that this critical carrier concentration is an intrinsic property of STO and determined by its electronic band structure.” in page 6.

- The authors assume that electrons are homogeneously distributed in the STO. I would expect that for the highest thicknesses of STO, one should recover the same situation than in LAO/STO heterostructures, with electrons confined at the interfaces. In this case, the low doping regime corresponding to the transition at $0.01e/Ti$ could be a pure artefact.

Response: Thanks very much for the comment. We agree with the reviewer that if the STO is thick enough, the scenario may recover to the one like that of the conventional LAO/STO heterostructure, due to the relaxation of the STO lattice and/or imperfect layer-by-layer growth of the STO layer. However, in the STO thickness range studied in this report, the low doping regime is still at the state where the strain from LAO holds, as supported by XRD measurement (Fig. S4) and the STEM strain mapping data (Fig. 2 and S5).

- The resistance vs temperature curves should be shown for the different samples. Do they exhibit the characteristic upturn in the resistance expected for WL?

Response: Thanks very much for the suggestion. As suggested, we plot the temperature dependent resistance of the samples below, Fig. R3, where significant upturns at low temperature were observed for all the structures. The inset of the figure shows the STO thickness dependent resistivity at 300 K, where the resistivity is higher with thicker STO layer. The feature is consistent with the modulation effect of the carrier concentration by the STO thickness. However, it should be noted that the upturn of the resistance at low temperature range might be suppressed with the presence of the spin orbit interaction theoretically, which is not the case here. A similar phenomenon was observed in topological insulator, which was ascribed to the factors that dominate the temperature dependent behaviour, but do not respond significantly to the external magnetic field [H. Z. Lu and S. Q. Shen, Phys. Rev. Lett. 112, 146601(2014), M. Liu *et al*, Phys. Rev. B 83, 165440 (2011)].

We have added the temperature dependent resistance in the revised supplementary materials as Fig. S8.

Fig. R3 | Temperature dependent sheet resistance of LAO//STO/LAO heterostructure as functions of the thickness of the STO layer. Inset shows the thickness dependent resistivity of the structures at 300 K.

- The authors should clearly indicate what is the exact formula used in the fit of magneto-resistance.
 Response: The supplementary material presents the derivation of the equation 1 in the main text. At the beginning, we used the equation 2 to carry out the fitting procedures on the magnetoresistance data. In order to present the magnetoconductance due to the weak anti-localization effect, we removed the classical Lorentz components obtained by the fitting procedures using equation 2, from the measured MR data. At last, the remained magnetoconductance and the fitting curves from the equation 1 with the corresponding fitting parameters are plotted in Fig. 5b.

To indicate procedures clearly, we added the sentence “which are obtained after removing the classical Lorentz components” in the first paragraph of page 13.

Reviewer #3 (Remarks to the Author):

- Key results: The paper of Lin et al. reports the study of the transport properties of LaAlO₃/SrTiO₃/LaAlO₃ heterostructures formed by two non-equivalent interfaces. They propose this system as a good candidate to study the spin-orbit Rashba effect, which is a really interesting property for the spintronic community. One of the main results given by this work is that, by varying the SrTiO₃ layer thickness, the authors achieved to tune the carrier density and observed experimentally a transition from a cubic to linear (more

precisely cubic+linear, if I understood correctly) Rashba effect, the linear Rashba effect being only predicted theoretically until now.

Response: Thanks very much for the reviewer's positive evaluation of our work.

- Validity: The manuscript is globally well written. Some precisions seem however necessary to judge about the complete validity of the paper.

Response: Thanks very much for the comment. In the revised manuscript, we have revised and improved the manuscript accordingly.

- Originality and significance: The observation of the linear Rashba effect will undoubtedly motivate further study to get a better understanding of this system and to get a better control of this interaction. It would thus be interesting to submit it for discussion. It is however not certain that this paper will get an audience out of the spintronic and oxide community.

Response: We agree with the reviewer that the observation reported in the manuscript is interesting for the spintronic and oxide community. Besides, we also expect that it will attract audience beyond this community because the spin-orbit interaction is a fundamental property of materials and it has been found to play important roles in various exotic materials ranging from layered graphene-like materials to cold atoms [A. Monchon *et al*, Nat. Mater. 14, 871 (2015), A. Soumyanarayanan *et al*, Nature 539, 509-517 (2016)].

- Data & methodology: The magneto-transport measurements are associated with some analysis of the atomic structure performed by STEM and some support from first-principles calculations.

For the DFT-based tight binding calculations, I would suggest to add more information about the parameters which were used and the methodology. In particular, it will be easier for the reader to have everything in hand than to collect all the information from previous references.

What is really from "first principles" and which parameters (energy splitting, atomic spin-orbit interaction, asymmetry strength coefficient) are chosen to fit with the experience? In Ref. 40, some parameters are obtained from DFT calculations, but performed on systems with different geometries and in-plane lattice parameters. Was it modified for the current paper?

L. 142-143, the authors say that the results are supported by DFT calculations and refer to the supplementary information. I did not see any detail about the DFT calculations (which code was used, which Exc functional...? (Wien2k + GGA?)).

How was the value given in Fig. S4d (purple square) obtained? With a slab geometry to simulate the effect of the internal electric field (with a STO thickness of 3-4 u.c.?), and an in-plane lattice parameter equal to the experimental or calculated one? How are defined the values represented by the dashed lines in the same figures (also calculations?).

Response: Thanks very much for the reviewer's comments and questions regarding the calculation. To address all these issues, we have provided more details regarding the method for the DFT-based tight binding calculations in the revised manuscript, page 15-16, "Density functional theory based tight binding calculations. To construct a realistic tight-binding model for STO based heterostructure and also avoid adjustable parameters, we performed a projection⁵⁴ of Wien2K⁵⁵ density functional theory (DFT) results for bulk STO onto maximally localized Wannier orbitals, which exactly reproduced the DFT calculated band structure^{40,56}. The tight binding Hamiltonian is described by $\mathbf{H}_o + \mathbf{H}_\xi + \mathbf{H}_\gamma$ in the $\mathbf{t}_{2g}(xy, yz, xz)$ basis. The \mathbf{H}_o term contains local energy terms ϵ and hopping terms t . In bulk STO with cubic symmetry $\epsilon_{xy} = \epsilon_{yz/xz}$, the compressive strain from the LAO substrate increases the xy orbital energy in the order of 10 meV. Without loss of generality, we set $\epsilon_{xy} - \epsilon_{yz/xz} = 20$ meV. There are three hopping terms: the large hopping term of $t_1 = 0.277$ eV arises from the large xy intraorbital hopping integral along the x and y directions; $t_2 = 0.031$ eV and $t_3 = 0.076$ eV indicate a much smaller hopping integral along the z and $(1,1,0)$ directions of the xy orbital, respectively. The \mathbf{H}_ξ term includes atomic spin-orbit interaction, whose strength is 19.2 meV, as estimated from the DFT calculated orbital splitting at Γ point. The last term $\mathbf{H}_\gamma = \langle xy | \mathbf{H} | yz/xz \rangle$ is an antisymmetric hopping between xy and yz/xz orbitals along the x/y direction. It describes the asymmetry due to the built-in electric field, which results in the Rashba spin splitting. In the bulk STO, \mathbf{H}_γ vanishes due to inversion symmetry, while at the LAO/STO interface \mathbf{H}_γ is 20 meV." We also added a description of how the STO lattice constant is calculated in the revised supplementary material (supplementary information 3).

About the chemical analysis, it is said that the two interfaces show clear cation intermixing, but that they remain non-equivalent. Was such analysis performed for different STO layer thicknesses? Or only for the 20 u.c.? From what is said, the out-of-plane STO layer lattice parameter is lower than expected mainly due to electrostrictive effect. Such effect should be linked with the asymmetry of the interfaces, and thus it should depend on the atomic structure at these interfaces. Did the authors notice any difference for the low and high layer thicknesses (that is, below and above 20 u.c., the thickness at which the transition occurs)?

Response: Thanks very much for the reviewer's comment. At this moment, we only did the STEM characterization for the structure with 20 uc STO. In this work, we did not focus on the electrostrictive effect because the transition of SOC occurs because of the modulation of the carriers introduced to the Ti site, which is an intrinsic property of the 3d orbitals of Ti in STO layer. In general, the imperfect interfaces provide a complementary way to achieve an energy gain to compensate the effect of the polar field from the LAO, which will influence the built-in field experienced by the STO layer. We agree with the reviewer that more systematic TEM investigations on the interfaces of the structure with different STO thickness may give us more information about the development of the built-in field in the STO layer, but this will require a large amount of experimental resources and time and we plan to devote a future project to this specific task.

The "rough" approximation which consists in averaging the number of electrons per Ti should be maybe discussed a little. The localization of the electrons near the interface is important as it depends on the internal electric field, which will itself have some impact on the Rashba splitting. Moreover, such approximation is quite contradictory with the fact that all the study relies on having two non-equivalent interfaces.

Response: Thanks very much for the comment. We agree with the reviewer that the calculated number of electrons per Ti is a "rough" approximation to estimate the doping level of the STO layer. As expected from the designed structure with two non-equivalent interfaces, the electrons distribution across the whole STO layer will show a gradual increase from the p-type interface to the n-type interface. The estimated electron number per Ti site gives only an average value, which does not reflect the local deviations along the STO layer. To give more discussions about this point, we have added the sentences "In the rough estimation, the carriers are assumed to be homogeneously doped in the entire STO layer, which does not reflect the local deviation of the electron filling across the STO layer. As expected from the designed LAO/STO/LAO structure with two non-equivalent interfaces, the electrons distribution across the whole

STO layer should show a gradual increase from the p-type interface to the n-type interface. Furthermore, the localized electrons near the interfaces may have an important influence on the internal electric field and impact on the Rashba splitting. Nevertheless, the carrier density estimated from the Hall effect measurement exhibits a consistent trend of dependence on the STO layer thickness.” in the first paragraph of page 9.

Is it possible to have some information on the method used to fit the parameters for Fig. S6? Above 20 u.c. the red and black points are superimposed, right? Why is the error increasing (of a factor 3) above 20 u.c.? This increase of the error is higher than the difference of errors between the fits with and without the linear contribution.

Response: Yes, the red and black points in Fig. S6 are superimposed for structures with above the 20 uc STO, indicating that both models give identical fitting results at this regime. The error bars are obtained as the total absolute values of the differences between the calculated and the measured conductivity. The larger errors for the structures with above 20 uc STO might be due to the larger component of the field independent conductivity, as seen from the fitting parameters in Table R1. However, as both models give almost the same errors and fitting parameters (Bso1 from the model including linear and cubic term is trivial for structures with above 20 uc STO), it will not influence the validity of our fitting results.

To give more information and discussion about the error analyses, we have added the sentences “To make the comparison quantitatively, we calculated the errors from both fitting models, which are obtained as the total absolute values of the differences between the calculated and the measured conductivity. As shown in Fig. S6c, the errors from the fitting with the linear Rashba term are smaller than that without the linear term, for the samples with STO thickness less than 20 uc. And for the samples with more than 20 uc STO, the fittings with and without the linear Rashba term result in almost similar error values.” in the end of the last paragraph of the page 11 in the revised supplementary materials.

In the supplementary paragraph S4, how are the effective mass determined?

Response: For the light band (the upper $d_{xz/yz}$ orbital), an effective mass of $1m_e$ is used because masses around $1m_e$ for the light band in LAO/STO interface were often extracted in previous reports [F. Trier *et al*, Phys. Rev. Lett. 117, 096804 (2016), Y. J. Chang *et al*, Phys. Rev. Lett. 111, 126401 (2013), A. D. Caviglia *et al*, Phys. Rev. Lett. 105, 236802 (2010)]. The calculated effective mass derived from the first-principle calculation in our previous work was directly used for the heavy band (the lower $d_{xz/yz}$ orbital) [Z. Zhong *et al*, Phys. Rev. B 87, 161102(R) (2013)].

Also, I have two questions for my understanding of the results:

1) If the carrier in STO are mainly here due to the polarity of LAO, can the authors explain a little more why the carrier density varies as a function of the STO thickness?

Response: Thanks very much for the question. The carrier distribution of carriers as a result of the polarity of LAO is inhomogeneous away from the STO/LAO or LAO/STO interface, and it is natural that the carrier density in the STO layers is dependent on the STO thickness if the layer is thin enough. Furthermore, as explained in ref. 35 of the manuscript, the forming energy of the polar-induced oxygen vacancies shows thickness (of both STO and LAO)-dependent behaviour. This indicates that the polarity effect of the LAO is dependent on the STO layer thickness too. Thus, we experimentally observed that the carrier density varies as a function of the STO thickness in the LAO//STO/LAO heterostructures.

2) Why do the authors finally show some evidence of the linear Rashba effects, while other did not? Is it linked with the high carrier density? The special geometry of the system? From the bandstructure, I can understand why the cubic Rashba is seen first, and the linear appears only after, but if we forget the band energy ordering, the linear Rashba should be also present for the standard LAO/STO interface (?). I did not get the hypothesis that the d_{xy} orbital being localized at the interface, the corresponding electrons would not participate to the charge transport.

Response: Thanks very much for the comment. Yes, we agree with the reviewer that the linear Rashba term should be present for the standard LAO/STO interface, however, which has not been reported so far. As shown in Fig. S1 for the band structure of the conventional LAO/STO heterostructures, the linear Rashba effect can appear at the lowest d_{xy} and the upper $d_{xz/yz}$ orbitals, and there are two Rashba transitions as the filling-electron numbers increase. According to the polar catastrophe scenario, there should be 0.5 e/Ti (around $3.1 \times 10^{14} \text{ cm}^{-2}$) transferred to the Ti site to avoid the accumulation of the potential. However, the carrier density is normally one order of magnitude smaller in the practical LAO/STO structures. Such discrepancy was ascribed to the localization of the d_{xy} electrons near the interface due to their confinement in the z direction [P. Delugas *et al*, Phys. Rev. Lett. 106, 166807 (2011)]. Therefore, in order to observe the linear Rashba effect, the doping level should be high enough to reach the upper $d_{xz/yz}$ orbital. The critical filling level is around 0.13 e/Ti, which equal $8.1 \times 10^{13} \text{ cm}^{-2}$. This carrier density is in the upper range of the experiments reported so far on investigating the Rashba effect (Ref. 25 and 42 of the manuscript). This might be the reason that the linear Rashba effect and the transition have not been reported so far.

- Conclusions: Regarding my previous comments, most of the conclusion are certainly reliable, but some of them would greatly benefit of more details (in particular about the methods).

Response: We thank the reviewer for the comments and questions, which help us further improve the manuscript. We hope that our replies and revised manuscript address the reviewer's concerns.

- References: ok

- Clarity and context: In the abstract, I would suggest to change “transition between cubic and linear” by “transition between cubic and linear+cubic”.

Response: Thanks very much for the suggestion. We have changed “transition between cubic and linear” by “transition between cubic and the coexistence of the linear and cubic Rashba effect” in the abstract.

- Particular part of the manuscript that you feel is outside the scope of your expertise: I am not expert in transport measurement and the full meaning of the ILP equations and their use is quite new for me. Sorry if some of my questions were "naive" in this sense.

Response: We truly appreciate the comments from the reviewer, which help us further improve the manuscript.

REVIEWERS' COMMENTS:

Reviewer #2 (Remarks to the Author):

The authors have considerably improved the quality of the manuscript by clarifying several points and adding complementary experimental results. They have also provided thorough and convincing answers to all reviewers' questions. To my opinion, the article can be published in Nature Communications without further modifications.

Reviewer #3 (Remarks to the Author):

Concerning the details of the calculations, I have some minor questions/comments:

1) Have the Wien2k calculations been done also within the PBEsol approximation? For a tetragonally distorted lattice? Is it normal (maybe yes) that $H\xi$ has almost the same value than in PRB 87 161102 (2013), which means it is independent on the strain? The effective masses are also considered for the undistorted system?

2) "We should note that this critical carrier concentration is an intrinsic property of STO and determined by its electronic band structure": is it not related to the choice of " $\epsilon_{xy} - \epsilon_{xz} + \epsilon_{yz}$ ", which will define the band occupation as a function of the position of the Fermi level? This band energy difference is chosen to 20 meV (instead of 10?) for the TB calculation, and it may also depend on the approximations in DFT calculations...?

3) For the VASP calculations, I would specify the pseudo-potentials which have been used. Do they include 3s3p semicores? Do the decrease of the c lattice parameter also appear if we do not add the oxygen vacancy? Is it really induced by a built in electric field or rather by the distortions induced by this defect? What would happen if we use a charged supercell?

If the easier formation of oxygen vacancies is also a good hypothesis to explain the variation of carrier density as a function of the STO thickness, maybe this should be put forward in the main paper.

I thank the authors to give all these details concerning their experiment and the calculations. I think it may help to check the validity but also to help people who would like to reproduce their methodology. I still think that some approximations are "rough" (as the authors acknowledged), but this is also related to the complexity of the system, and as long as it is correctly mentioned in the manuscript, this should not prevent a publication. I thus recommend it for publication.

Response to the referees' comments

(The referee's comments are quoted and our point-to-point responses in blue. Changes made in the revised manuscript are marked in blue.)

Reviewer #2 (Remarks to the Author):

The authors have considerably improved the quality of the manuscript by clarifying several points and adding complementary experimental results. They have also provided thorough and convincing answers to all reviewers' questions. To my opinion, the article can be published in Nature Communications without further modifications.

Response: We are glad that our revised manuscript have addressed the concerns of the reviewer and thank his/her supportive conclusion for our work.

Reviewer #3 (Remarks to the Author):

Concerning the details of the calculations, I have some minor questions/comments:

1) Have the Wien2k calculations been done also within the PBEsol approximation? For a tetragonally distorted lattice? Is it normal (maybe yes) that H_{ξ} has almost the same value than in PRB 87 161102 (2013), which means it is independent on the strain? The effective masses are also considered for the undistorted system?

Response: Answered together with question 2)

2) "We should note that this critical carrier concentration is an intrinsic property of STO and determined by its electronic band structure": is it not related to the choice of " $\epsilon_{xy} - \epsilon_{xz+yz}$ ", which will define the band occupation as a function of the position of the Fermi level? This band energy difference is chosen to 20 meV (instead of 10?) for the TB calculation, and it may also depend on the approximations in DFT calculations...?

Response: The Wien2k calculations have been done within PBE potential for cubic structure; " $\epsilon_{xy} - \epsilon_{xz+yz}$ " is defined with respect to Fermi energy level. Structural distortion and computational details will moderately modify the hopping terms t (effective mass) and crystal field splitting ($\epsilon_{xy} - \epsilon_{xz+yz}$), but not change the atomic spin orbit interaction term H_{ξ} . The Rashba transition as well as the corresponding critical carrier concentration mainly depend on H_{ξ} and the t_{2g} band structure character of STO. We have checked that a change of hopping terms t and crystal field splitting will not influence our conclusion. We added the discussion into the method part marked in blue.

3) For the VASP calculations, I would specify the pseudo-potentials which have been used.

Do they include 3s3p semicores? Do the decrease of the c lattice parameter also appear if we do not add the oxygen vacancy? Is it really induced by a built in electric field or rather by the distortions induced by this defect? What would happen if we use a charged supercell?

Response: We used the supplied VASP PAW pseudopotentials (La_s, Sr_sv, Al Ti_pv and O) with the 5p65d16s2 valence configuration for La, 4s24p65s2 for Sr_sv, 3s23p1 for Al, 3p63d34s1 for Ti_pv, and 2s22p4 for O. The decrease of c lattice parameter will disappear without the formation of oxygen vacancy. Nevertheless, we have also studied bulk STO with an oxygen vacancy, and do not find the decrease of lattice constant, which indicates that the electrostriction effect in LAO/STO is induced by the electric field but not the distortion induced by oxygen vacancies. A charged supercell shows similar behaviour.

If the easier formation of oxygen vacancies is also a good hypothesis to explain the variation of carrier density as a function of the STO thickness, maybe this should be put forward in the main paper.

Response: Thanks for the suggestion. We have added the sentence “which may be achieved by influencing the formation energy of the oxygen vacancy” at the first paragraph of the page 10.

I thank the authors to give all these details concerning their experiment and the calculations. I think it may help to check the validity but also to help people who would like to reproduce their methodology. I still think that some approximations are “rough” (as the authors acknowledged), but this is also related to the complexity of the system, and as long as it is correctly mentioned in the manuscript, this should not prevent a publication. I thus recommend it for publication.

Response: We thank the comments and suggestions given by the reviewer, which help us further improve the quality of the manuscript. In addition, we thank his/her recommendation for the manuscript’s publication.